# Ethical Decision-Making Guidelines for Mental Health Clinicians in the Artificial Intelligence (AI) Era

**DOI:** 10.3390/healthcare13233057

**Published:** 2025-11-25

**Authors:** Yegan Pillay

**Affiliations:** Department of Counseling and Higher Education, Ohio University, Athens, OH 45701, USA; pillay@ohio.edu; Tel.: +1-(740)-593-1000

**Keywords:** artificial intelligence, ethics, mental health

## Abstract

The meteoric rise in generative AI has created both opportunities and ethical challenges for the mental health disciplines, namely in clinical mental health counseling, psychology, psychiatry, and social work. While these disciplines have been grounded in well-established ethical principles such as autonomy, beneficence, justice, fidelity, and confidentiality, the exponential ubiquity of AI in society has rendered mental health professionals unsure as to how to navigate ethical decision making in the AI era. The author proposes a preliminary ethical framework which synthesizes the code of ethics of the American Counseling Association (ACA), the American Psychological Association (APA), the American Medical Association (AMA), and the National Association of Social Workers (NASW), which is then organized around five pillars: (i) autonomy and informed consent; (ii) beneficence and non-malfeasance; (iii) confidentiality, privacy, and transparency; (iv) justice, fairness and inclusiveness; and (v) fidelity, professional integrity, and accountability. These pillars are juxtaposed with AI ethical guidelines developed by multinational organizations, governmental and non-governmental entities, and technology corporations. The resulting integrated ethical framework provides a practical cogent structure that mental health professionals can use when navigating this uncharted terrain. A case study based on the proposed ethical framework and strategies that clinical mental professionals can consider prior to incorporating AI into their clinical repertoire are offered. Limitations of the framework and its implications for future research are addressed.

## 1. Introduction

Generative artificial intelligence (AI) as it is known in contemporary society can be credited to a series of steps beginning in 1822 with the invention of the computer by Charles Babbage and in the twentieth century by the work of Alan Turing, the British mathematician, computer scientist, and military officer who is considered to be the father of computer science and AI (Grzybowski, et al., 2024) [1]. Although AI has been around for decades, the 21st century has seen an exponential growth in the attention and utilization of generative AI. According to Blinko (2025) [2], ChatGPT, which was developed by Open AI, had one million users in the first five days of its launch in 2022; 100 million weekly users in 2023; and approximately 400 million users in 2025.

The Oliver Wyman Forum [3] conducted two generative AI-specific surveys in 2023 with a sample of 25,000 respondents across 16 countries, namely Australia, Brazil, Canada, China, France, Germany, Hong Kong, India, Indonesia, Italy, Mexico, Singapore, Spain, the United Arab Emirates, the United Kingdom, and the United States. One of the survey items asked participants (N = 16,033) to respond to the question “Which of the following areas do you think AI will help improve most in the next 30 years?” The findings were as follows: health care (41%), transportation (35%), quantum computing (32%), education (32%), media and entertainment (29%), environmental conservation (28%), and energy (25%).

While general health care ranked the highest among the various categories, a notable finding was that 77% of the participants who had never sought mental health services previously were willing to try generative AI therapy. Moreover, statistical modeling projections by the Oliver Wyman Forum [3] suggest that generative AI will increase access to 400 million mental health patients worldwide by the year 2030. While these forecasts appear promising for the provision of mental health services globally, uncertainty remains as to whether AI will transform society positively or lead to negative mental-health-related ethical outcomes.

The ethical guidelines for the preeminent mental health disciplines are well established with the first code of ethics published for psychologists by the American Psychological Association in 1953, with revisions in 1977, 1992, 2002, 2010, and 2016 [4]. This was followed by the adoption of the first ethical guidelines for mental health counselors by the American Personnel and Guidance Association, the predecessor to the American Counseling Association, with revisions in 1981, 1995, 2004, and 2014 [5]. The current version addressed ethical guidelines related to social media, multiculturalism, and diagnosis. The ethical guidelines for social workers were developed in 1979 by the National Association of Social Workers (NASW), with revisions in 1996, 2008, 2013, and 2021 [6]. The current version updated the standards on technology, client privacy, and documentation. In 1973, the American Psychiatric Association published ‘The Principles of Medical Ethics’, which was based on the American Medical Association code of medical ethics that focused on the psychiatric context, and it was updated in 2013 to include confidentiality, boundary violations, and involuntary treatment [7].

It is evident that the various relevant mental health disciplines’ ethical codes of practice have a long history of periodic revisions. Although there is no single ethical code for generative AI, ethical guidelines have recently been formulated by multinational organizations, governments, technology corporations, and academic and non-academic entities. The challenge, however, is a cogent set of ethical principles that specifically guide the ethical responsibility of mental health professionals who are navigating the potential of incorporation of AI into their clinical practice. To address this current gap in the literature, the author utilized narrative review methodology, summarized in the section below, to synthesize and integrate the established core ethical principles of the ACA, APA, NASW, and AMA [4,5,6,7] with the emerging ethical principles of multinational organizations, governmental and non-governmental entities, and technology corporations, with the objective of conducting conceptual exploration and advancing a synthesized set of guidelines that mental health clinicians across disciplines can consult to guide AI-related decision making.

## 2. Methodology

Narrative reviews ‘provide a readable, thoughtful, and practical analysis on a topic…allow review authors to advance new ideas while describing and interpreting literature in the field” (Sukhera, 2022, p. 416) [8]; thus, a narrative review was selected by the author as the research methodology of choice to guide the development of the ethical framework. The steps in conducting a narrative review as articulated by Snyder (2019) [9], Ferrari (2015) [10], Toracco (2005) [11], and Baumeister & Leary (1997) [12] were sequentially performed. They include defining a purpose, planning a search strategy, selecting relevant literature, extracting and synthesizing the data, and developing a framework. The purpose and the scope of the narrative review was informed by the question, “Do the ethical guidelines of the professional mental health disciplines, government entities and multinational technology organizations adequately address the ethical use of AI in mental health practice?” The goal was to address the conceptual gap in the current literature and propose an interdisciplinary cogent ethical framework.

The search engines PsychINFO, PubMed, Eric, and Google Scholar were used to conduct a review of the literature during May–August 2025 using the following keywords and Boolean operations: “ethical decision making guidelines AND artificial intelligence AND mental health clinicians”; “counseling code of ethics, psychology code of ethics, psychiatry code of ethics, social work code of ethics AND technology code of ethics”; “non-governmental, governmental, multinational companies AND artificial intelligence AND/OR ethics”. Peer reviewed empirical and conceptual publications and gray literature (e.g., reports and policy documents of governments, non-governmental and multi-national technology organizations) were reviewed to ensure that the latest developments related to the ethical decision-making guidelines were comprehensively addressed. The inclusion criteria for a literature review comprised the ethical principles of the American Counseling Association (ACA, 2014) [5], the American Psychological Association (APA, 2017) [4], the American Medical Association (AMA, n.d.) [7], and the National Association of Social Workers (NASW, 2021) [6]. These four professional organization codes were selected because they represent the most dominant ethical practices that regulate mental health in the US and have relevance globally. In addition, the ethics governing AI advocated by multinational organizations such as the Organization for Economic Co-operation and Development. (2019) [13] and UNESCO (2021) [14]; by governments such as the US Department of Defense (2020) [15] and the UK Government (2020) [16]; by technology corporations such as Google (2018) [17], Microsoft (n.d) [18], and IBM (2018) [19]; and academic and non-academic entities such as The Future of Life Institute (2017) [20], The Montreal Declaration (2018) [21], IEEE (2019) [22], and the European Commission High Level Expert Group on Artificial Intelligence (2019) [23] were reviewed. These sources were selected because they explicitly outlined the ethical principles of the mainstream mental health disciplines, government and non-governmental entities, and technology companies. The ethical guidelines of the four professional mental health disciplines that were unrelated to mental health, technology, or AI and AI guidelines of governmental, non-governmental, and technology corporations that were not applicable to mental health were excluded in the data extraction process. The extraction process included examining the ethical principles of the core ethical guidelines of the ACA, APA, AMA, and NASW [4,5,6,7], namely autonomy and informed consent; beneficence and non-malfeasance; confidentiality, privacy, and transparency; justice, fairness, and inclusiveness; fidelity, professional integrity, and accountability; and the ethical guidelines for AI by multinational technology corporations and government and non-government stakeholders. The ethical principles of the respective entities were synthesized and reorganized relative to convergent themes and mapped into the existing ethical domains of the professional mental health associations. This process led to the distilling of the integrated mental health and AI ethical principles for mental health professionals.

## 3. Ethical Principles of Mental Health and AI

### 3.1. Autonomy and Informed Consent

The mental health disciplines emphasize the client’s right to self-determination, respect for their dignity, and their right to make choices (ACA, 2014, Section A.1.a; APA 2017, Principle E; AMA, n.d., Opinion 1.1.3 and NASW, 2021 Section 1.02) [4,5,6,7]. Clinicians are required to use culturally and developmentally appropriate language to obtain informed consent when conducting assessment, research, teaching, and treatment, which includes the goals and techniques, the risks and benefits of therapy, an exploration of alternatives, and the recognition that informed consent is ongoing and context-specific (ACA, 2014 A.2.a.; APA, 2017, Standard 3.10; AMA, 2.1.1; NASW, 2021 Section 1.03) [4,5,6,7]. According to the IEEE (2019) [22] and the OECD (2019) [13], AI ethics guidelines indicate that human decision making and human oversight must be preserved and supported. Moreover, AI systems ought to be designed in such a way that they use language that is understood and explainable so that clients are aware when they are interacting with AI systems so that they can make an informed decision about how their data will be used, stored, and transmitted. An important caveat is that the consenting process is performed without coercion, and the client can revoke consent at any time.

### 3.2. Beneficence and Non-Malfeasance

Beneficence and non-malfeasance are ethical principles that guide mental health professionals to promote the well-being of clients while simultaneously abiding by the duty to cause no harm. The ethical codes of the major mental health disciplines address promoting individual well-being, growth, and advocacy for vulnerable populations while simultaneously minimizing treatment interventions or the abuse of power that may cause harm (ACA, 2014 A.1.a, A.4.a; APA, 2017 Principle A; AMA, n.d. Principles, I & VIII; NASW, 2017 1.01) [4,5,6,7]. Similarly, AI entities such as IEEE (2019), OECD (2019), IEEE (2019), IBM (2018, Google (2018), and the Future of Life Institute (2017) [13,17,19,22] advocate to enhance the well-being of individuals and society and prevent any unintended harm that may occur. The recommendation by UNESCO (2021) [14], “AI systems must not harm human beings, either through their design or their implementation. The ‘do no harm’ principle must be upheld as a core value” (https://www.unesco.org/en/artificial-intelligence/recommendation-ethics (accessed on 31 May 2025)), encapsulates the sentiment of AI entities.

### 3.3. Confidentiality, Privacy, and Transparency

Ensuring confidentiality, privacy, and transparency by mental health and AI professionals are prerequisites for informed consent and the development of trust. Mental health professionals are required to protect all client information except in the case of ethical or legal justification (ACA, 2014, B.1.b.-B.4.b.; APA, 2017, Standards 4.01–4.07; AMA, n.d. Opinions 3.1.1–3.1.2; NASW, 2021, 1.07) [4,5,6,7]. Mental health professionals ought to allow their clients to control what information they share and with whom the information is shared and clarify confidentiality and its limits. Moreover, the purpose of data collection, especially when AI or digital tools are used in the diagnosis and treatment, ought to be addressed as part of the consent process (ACA, 2014, Section B; Section H.; APA, 2017, Principle E; AMA, n.d., Opinion 3.3.2; NASW, 2021, 1.07 a-n) [4,5,6,7]. The ethical codes by IEEE (2019) and OECD (2019) [13,22] require that AI systems for mental health, health care, and social services must ensure that sensitive information, e.g., HIPAA, is inaccessible to unauthorized parties, is processed with stringent security measures, respects the user’s autonomy by obtaining informed consent regarding what data can be collected, ensures that AI systems are understandable and explainable, ensures that users are apprised when data is being collected using AI, and explains how decisions are being made for health-related issues.

### 3.4. Justice, Fairness, and Inclusiveness

Mental health counselors, psychologists, psychiatrists, and social workers by virtue of the ethical principles that guide their practice are expected to avoid discrimination and treat their clients in a manner that promotes equitable access, social justice, and advocacy for marginalized groups in clinical practice, research, and training in the context of social determinants of health (ACA, 2014, Preamble, Section A.2.c, C.5, F. 11.c; APA, 2017, Principle D, Ethical Standards 3.01, 9.06; AMA, n.d., Opinion 1.1.2; NASW, 2021 [4,5,6,7], Preamble, Section 1.05, 6.01, 6.04) [6]. The ethical guidelines that govern international, government entities and corporate, academic, and non-profit organizations such as OECD (2019), UNESCO (2021), IEEE (2019) [13,14,22], etc., address bias mitigation and promote inclusivity and access to all non-discrimination policies that promote user engagement in decision making when using AI.

### 3.5. Fidelity, Professional Integrity, and Accountability

The principles of commitment to clients of mental health services and other professionals, taking responsibility for one’s decisions and actions, and adherence to ethical and moral standards are the building blocks for the establishment of trust in the counseling, psychology, psychiatry, and social work disciplines. With the ambiguity and the growth around generative AI, it is imperative that trust is firmly established. Mental health professionals are required to demonstrate the qualities of trustworthiness, avoid deception, report unethical behavior, and seek consultation and supervision, when necessary, in teaching, research, and practice (ACA, 2014, Preamble, C.3.a.; APA, 2017, Principle C; AMA, n.d., Principle II, Opinion 11.2.7; NASW, 2021, Preamble, 1.06.4.01) [4,5,6,7]. Mental health professionals are required to represent their qualifications accurately and maintain competence (ACA, 2014, C.4.a-f; APA, 2017, 2.03; AMA, n.d., Principle V; NASW, 2021, 4.01) [4,5,6,7]. The ethical principles of AI as outlined by OECD (2019) and IEEE (2019) [13,22] direct AI developers to take into consideration human values and ensure trustworthiness and reliability in combination with creating systems that are transparent, understandable, and fair in both the design and how it is deployed. Moreover, AI ethical guidelines highlight the imperative that AI developers ensure accountability through a design that comprises audit, human oversight, and provides recourse when harm or an error occurs.

The ethical principles guiding mental health professions—counseling, psychology, psychiatry, and social work—have been a dynamic set of parameters that have been developed over an extended period to protect the client considering emerging societal and technological changes. Ethical principles emphasize autonomy and informed consent; beneficence and non-malfeasance, confidentiality and transparency; justice, fairness, and inclusiveness; and fidelity, professional integrity, and accountability. Similarly, with the meteoric rise of generative AI, new ethical challenges have emerged, prompting global organizations, government entities, and corporations to develop complementary AI ethics guidelines that stress human-centric values, transparency, explainability, fairness, non-discrimination, privacy, data protection, accountability, robustness, sustainability, social good, and human oversight. The convergence of these frameworks underscores the need for comprehensive ethical guidance to help mental health professionals navigate the integration of AI in clinical, research, and educational settings while safeguarding client rights and well-being. In the sections that follow, an ethical framework specifically for mental health professionals is proposed.

## 4. Ethical Framework for Mental Health Professionals

The proposed practice-oriented clinical guidelines that emerged from this iterative narrative review process, which are summarized below, examine the intersection of established mental health ethical principles and the recently developed generative AI guidelines.

### 4.1. Autonomy and Informed Consent

i.Clinicians must disclose to the client whenever AI is used in their treatment, and this disclosure must include AI’s capabilities, limitations, potential impacts on their diagnosis, access to treatment, and cost implications.ii.Clinicians must provide information regarding the type of AI tools that will be used, their impact on the client’s treatment, how the data are collected, stored, and analyzed, and the role and involvement of third parties in the process if relevant.iii.Clinicians must be willing to allow the client to exercise their right to opt out of assisted AI treatments or decision-making processes and where feasible offer a human-based alternative.iv.Clinicians must ensure that the language used provides clear and understandable details about the use of AI to empower clients to give consent that is fully informed.

### 4.2. Beneficence and Non-Malfeasance

i.The therapeutic relationship remains central to ethical clinical care, and therefore the use of AI must not be viewed as a substitute for human connection but rather as a complementary modality that enhances the therapeutic alliance in assessment, diagnosis, and treatment planning.ii.AI must only be explored as a complementary modality when the clinician determines that it is competent in understanding, interpreting, and explaining its results to the client and relevant stakeholders.iii.Clinicians must select AI tools that are culturally appropriate to minimize the perpetuation of inequities and are reliable, valid, and have evidence-based research support.iv.Clinicians must use AI tools to promote client well-being and minimize risk rather than justifying or contributing to discriminatory practices, which should include algorithms to audit, identify, and address ethical concerns or unintended risk.v.AI tools must adhere to professional and ethical standards and must be evaluated for accuracy and appropriateness on a regularly scheduled basis.

### 4.3. Confidentiality and Transparency

i.The same confidentiality standards, e.g., compliance with HIPAA, federal and state laws, and relevant professional codes including secure data practices, ethical recordkeeping, encryption, and storage protection, must also apply to AI tools.ii.Clinicians must ensure compliance with confidentiality, privacy, and ethical standards, including HIPAA and relevant professional and legal regulations when third-party vendors or AI platforms are used.iii.Clinicians must be ethically accountable for the use of AI tools and must provide accurate information about AI capabilities and limitations, avoiding misleading claims.iv.Clinicians should advocate transparency by disclosing how AI models are developed and how algorithms are applied.

### 4.4. Justice, Fairness, and Inclusiveness

i.Clinicians have a responsibility to ensure that AI systems do not disadvantage individuals or justify discrimination based on marginalized identities.ii.Clinicians must examine the AI system’s function, design, and output to ensure equitable and ethical application and evaluate assessments for cultural bias and fairness.iii.Clinicians must promote AI systems that enhance justice and safety for all clients and must advocate for inclusive and transparent design features that support ethical obligations of mental health professionals.

### 4.5. Fidelity, Professional Integrity, and Accountability

i.Clinicians should only use AI tools when they have the training and competence to interpret the results responsibly and accurately and fully understand the implications, limitations, and the ethical use of AI.ii.Clinicians must stay informed regarding the best practices and risks regarding emerging tools by staying current with training in AI and digital ethics, in collaboration with AI technologists.iii.Clinical supervisors must model ethical AI use and guide supervisees in critically evaluating algorithmic tools in their clinical practice.

In sum, the comprehensive narrative review included the author mapping the convergent themes into five pillars of ethical principles, which were then synthesized thorough an iterative process into clinical practice guidelines. The application of the guidelines is illustrated in the hypothetical case study that follows.

## 5. Case Study

Background:

Dr. Ozan Emre is licensed as a professional supervising counselor and a professor who does pro bono work at a private behavioral health clinic serving a diverse urban population. The clinic recently adopted an AI-assisted platform, *Wysa*, (Premium Version) which is designed to combine chatbot self-help (CBT, DBT, mindfulness) and hybrid clinician/AI models to support clients as a between-session tool.

Client Profile: Undocumented Muslim refugee from Afghanistan, employed as an Uber driver.

Name: Abdi K.

Age: 35.

Presenting Concerns: Generalized anxiety, work related stress, PTSD.

Treatment Plan: Cognitive Behavioral Therapy (CBT) and Eye-Movement Desensitizing and Reprocessing (EMDR).

### 5.1. Application of the Ethical Framework

#### 5.1.1. Autonomy and Informed Consent

Dr. Emre provides information about *Wysa* in a clear and coherent manner that takes into account Abdi’s English language proficiency and explains the following:i.The purpose of using the AI tool *Wysa*, i.e., to support the client between sessions.ii.The benefits and the limits of *Wysa*, emphasizing that the tool does not replace Dr. Emre’s role as a clinician.iii.Assurance that the information collected will be analyzed and stored securely and information of the extent to which the vendor *Wysa* has access to data for maintenance purposes.iv.Abdi has the option to opt out of AI support and continue with traditional therapy.v.If Abdi provides consent to use the AI tool, he has the option to revoke consent at any time.

#### 5.1.2. Beneficence and Non-Malfeasance

Dr. Emre emphasizes the benefits of the AI tool *Wysa*, specifically that it does not replace the role of the therapist and the human therapeutic relationship but rather is an additional support mechanism in the absence of the therapist between sessions. To ensure that the incorporation of *Wysa* does not harm the client, Dr. Emre ought to undertake the following steps:i.Complete training and professional development relevant to AI ethics and the interpretation of data that is specific to the *Wysa* platform.ii.Focus on rapport building with Abdi.iii.Review any notes generated by *Wysa* for cultural relevance and accuracy.iv.Ensure that no clinical recommendations are implemented without his or other clinical staff review.v.Review scientific evidence for *Wysa*’s validity and reliability and efficacy for multicultural and international populations.

#### 5.1.3. Confidentiality and Transparency

Dr. Emre documents in Abdi’s clinical notes that *Wysa* is being used as part of the treatment protocol, the procedure related to the consenting process, and what precautions have been taken to ensure data security. Dr. Emre ensures HIPAA compliance by undertaking the following steps:i.Informing Abdi who can have access to his AI data and for what reasons.ii.Restricting how AI data is shared and with whom and that the data is encrypted and stored on secure servers.iii.Discussing that not all of AI’s functions are transparent such as algorithms, which are the property of the vendor, but that every step has been taken to ensure the credibility, fairness, and security of Wysa as an AI platform.

#### 5.1.4. Justice, Fairness, and Inclusiveness

Dr. Emre acknowledges that he has an ethical responsibility to ensure that Abdi is not excluded or treated unfairly because of his national origin or religious affiliation and is not disadvantaged by using *Wysa* as an AI tool. He accomplishes this by undertaking the following:i.Verifying that *Wysa* as a generative AI tool has been trained on various linguistic datasets so that bias could be reduced.ii.Reporting bias to the vendor if he notices frequent errors that may be related to marginalized groups and advocates for regular equity audits for how Wysa may be performing across different demographic groups, ethnicities, and races.

#### 5.1.5. Fidelity, Professional Integrity, and Accountability

Dr. Emre maintains professional integrity by accomplishing the following:i.Staying current with AI developments through conference attendance and other continuing professional development and by completing training related to AI/digital ethics, and in the case of Abdi with the use of the *Wysa* AI platform.ii.As a mental health professional and as a clinical supervisor, Dr. Emre ought to model the ethical use of AI, especially when training future mental health professionals to critically evaluate and not be reliant on algorithmic outputs so that the client is not harmed by the inclusion of AI in the clinical setting.iii.Dr. Emre ought to also communicate when AI outputs ought not to be used and be transparent in his rationale.

Based on the extant review and synthesis of the selected ethical principles, the author offers an additional checklist that mental health clinicians can consider prior to considering embarking on incorporating AI into their clinical practice repertoire.

## 6. Checklist for Clinicians Prior to AI Use

i.Consent: Convey to the client what AI does, its limitations, how data will be managed or used, and allowing the option to opt out in lieu of offering a human option alternative.ii.Vett vendors: The AI platform ought to be tested and peer reviewed for efficacy or biases prior to its adoption.iii.Data security: Ensure that data is encrypted and stored securely and explained clearly to the client regarding how it will be used. The specifics of a Business Associate Agreement ought to be discussed in the consenting process.iv.Competence: Ensure that you have sufficient knowledge to interpret the AI outputs and understand the scope of the algorithms.v.Inclusivity: Evaluate whether the AI platform has been designed for diverse populations.vi.Transparency: Be clear with clients about what AI does and not do and remain accountable for the decisions made.

## 7. Limitations and Future Directions

This seminal paper offers a preliminary exploration of a framework that provides an ethical structure for mental health professionals navigating the integration of generative AI into their practice, with the caveat that several limitations must be acknowledged. First, generative AI is a rapidly evolving field as is evident in its meteoric rise. Therefore, the proposed framework may have to go through several iterations as changes in tools and systems occur and the potential for accompanying ethical concerns emerge. It is recommended that feedback is solicited from the various interdisciplinary stakeholders and professional peers to settle on a set of ethical guidelines upon which consensus is reached. Secondly, while efforts in this paper were to consolidate human-related ethical guidelines with machine-related ethical guidelines, this amalgamation may be fraught with practical real-world implications and applications. Specifically, mental health clinicians may not have the technological knowledge to interpret or understand AI outputs or to explain to the client the scope of AI algorithms. It is recommended that AI vendors provide comprehensive and detailed information in layperson language so that mental health clinicians could be competent in providing salient information that is clearly understood by the client. Clinicians ought to seek opportunities for continuing education as it relates to specific AI platforms that they may have adopted. Thirdly, the relative infancy of AI as a complementary clinical tool has resulted in gaps in the curricula of clinical mental health clinician training programs, which has rendered clinical mental clinicians inadequately prepared to integrate AI into their clinical repertoire. The status quo may have a negative influence on the clinician’s willingness to integrate AI into their clinical practice and hence relate to the ethical guidelines as articulated in this manuscript. It is recommended that accrediting bodies and institutions of higher learning explore and prepare their graduates to navigate and be relevant in the rapidly changing AI landscape. Fourthly, the proposed guidelines have been premised on dated, Western-based, Euro-centric ethical codes. It is recommended that future iterations examine the ethical principles of relevant organizations that represent the current diverse global community. Finally, the proposed framework has not been empirically tested and validated in clinical settings and is limited by the narrative review methodological design. It is recommended that future researchers explore other research designs such as conducting a scoping review or using data collection methods such as interviews, focus groups, and surveys of clinicians and professional peers to provide empirical validation of the proposed framework.

## 8. Conclusions

It is evident that AI has already begun to reshape the societal landscape. If future mathematical projections come to fruition, such as those by the Oliver Wyman Forum, that 40 million more individuals globally will have access to mental health service, it poses both an exciting opportunity but also adds ethical complexities. While the established flagships for mental health disciplines such as clinical mental health counseling, psychology, psychiatry, and social work have long established ethical guard rails in place, clinical mental health professionals will be traversing uncharted territory in navigating and integrating these established ethical principles with the AI ethical principles advocated by international organizations, governments, technology corporations, and academic and non-academic entities. The author advocates that rather than seeing AI as a threat, mental health stakeholders ought to explore the opportunities that AI may present, with the caveat that they maintain their long-standing ethical compass and identity as mental health professionals while embracing the strengths of generative AI through principled and measured ethical lenses.

To this end, the proposed framework has accentuated the core ethical principles that form the foundation of our ethical practice as mental health professionals, which remains crucially relevant in guiding and navigating our AI integration, namely autonomy, informed consent, beneficence, non-malfeasance, justice, fairness, inclusiveness, fidelity, professional integrity, accountability, confidentiality, privacy, and transparency. Juxtaposed within each of the core ethical principles, the author has weaved in the relevant ethical AI guidelines that resulted in the proposed decision-making framework for adapting to emerging technological realities while simultaneously placing the clinician and the client at the center and being advocates for ethical and responsible integration of AI in our respective disciplines.

## Data Availability

No new data were created or analyzed in this study.

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
