# Peer review of "Ethical Decision-Making Guidelines for Mental Health Clinicians in the Artificial Intelligence (AI) Era"

_healthcare, 2025, doi:10.3390/healthcare13233057_

Round 1
Reviewer 1 Report
Comments and Suggestions for Authors
Dear editor,
Thank you for the opportunity to review this paper. I consider the paper interesting and part of an important endeavor. to think about ethical aspects of the use of new and emerging technologies, not the least those currently termed AI.
Some specific comments:
Lines 50–51:
“a notable finding was that 77% (one in three respondents)”. This should be looked at again – what does the author mean here?
Section 2: “2. Ethical Principles of Mental Health and AI”:
It seems that the perspective taken in this section is largely an American one. At least, this should be acknowledged by the author. Typically, when a paper takes the perspective of a single country as a case fr an article, this is acknowledged clearly in the paper.
Another possible route is that perhaps the author would like to add an international perspective with more references to policies from outside of the U.S.?
Line 106:
Reads: “are required to use culturally and developmental appropriate language”,
Should read: “are required to use culturally and developmentally appropriate language”
Lines 108–9:
Reads: “an exploration alternatives ”
Should read: “an exploration of alternatives ”
Lines 113–116:
The author writes: “Moreover, AI systems ought to be designed in such a way that they use language that is 113
understood and explainable so that consumers are aware when they are interacting with 114
AI systems so that they can make an informed decision about how their data will be used, 115
stored and transmitted.”
– But why is the terminology of a “consumer” used? Oftentimes, users of such tools are not consumers, since the tools might very well be free for the users but instead monetize their usage in other ways, and this should perhaps be acknowledged by the authors choice of words?
Section 3 “Ethical Framework for Mental Health Professionals”
The author writes that “The proposed framework for mental health professionals when navigating the complexities of AI draws from the intersection of the established mental health ethical codes and the recently developed generative AI guidelines.”
But it is not clearly stated in relation to the results exactly what is taken from where. Five principles are detailed, but it is not clear how these have come about, and in what sense they are in the intersection between the two domains. That is left to the reader to figure out. I suggest that the author provides some more guidance to the reader in this regard.
Section 4. “Limitations”
Lines 267–270:
The author writes: “Secondly, while efforts in this paper were to consolidate human related ethical guidelines with machine related ethical guidelines, this amalgamation may be fraught with practical real-world implications and applications.”
Here one might wonder. What, more precisely, might this be? I think the paper would benefit from a bit more clarity here.
Lines 274–275:
The author writes: “The proposed framework has not been empirically tested and validated in clinical settings”
This statement is interesting. What would such a validation look like? The framework consists (mostly) of a series of imperatives – how does the author conceive of them being validated? A few words on this might shed light on the matter.
Finally, a small general remark: I think that the text is quite full of unintentional double spaces, which can be removed using the “Find and replace” feature of most word processors.
Author Response
Reviewer Comment #1:
Lines 50–51:
“a notable finding was that 77% (one in three respondents)”. This should be looked at again – what does the author mean here?
Thank you --- this sentence has been edited
Reviewer Comment #2:
Section 2: “2. Ethical Principles of Mental Health and AI”:
It seems that the perspective taken in this section is largely an American one. At least, this should be acknowledged by the author. Typically, when a paper takes the perspective of a single country as a case fr an article, this is acknowledged clearly in the paper.
Another possible route is that perhaps the author would like to add an international perspective with more references to policies from outside of the U.S.?
This is a relevant comment. The methodology section has been revised since the first submission to address that the ethical codes (for practical purposes) as advocated in the literature (new literature citations were added) were delimited to 5 professional organization. This delimitation is addressed in the limitations section. Based on this feedback and the feedback from other reviewers two sections were added (a hypothetical case and tips for considering AI in clinical practice) was added to the manuscript to illustrate the application of the ethical framework and specifically using in the context of the international client and mental health professionals.
Reviewers Comment # 3 & 4 :
Line 106:
Reads: “are required to use culturally and developmental appropriate language”,
Should read: “are required to use culturally and developmentally appropriate language”
Lines 108–9:
Reads: “an exploration alternatives ”
Should read: “an exploration of alternatives ”
Lines 113–116:
The author writes: “Moreover, AI systems ought to be designed in such a way that they use language that is
understood and explainable so that consumers are aware when they are interacting with 114
AI systems so that they can make an informed decision about how their data will be used, 115
stored and transmitted.”
– But why is the terminology of a “consumer” used? Oftentimes, users of such tools are not consumers, since the tools might very well be free for the users but instead monetize their usage in other ways, and this should perhaps be acknowledged by the authors choice of words?
Thank you-- all edits suggested above completed based on the comments
Reviewer Comment # 5
The author writes that “The proposed framework for mental health professionals when navigating the complexities of AI draws from the intersection of the established mental health ethical codes and the recently developed generative AI guidelines.”
But it is not clearly stated in relation to the results exactly what is taken from where. Five principles are detailed, but it is not clear how these have come about, and in what sense they are in the intersection between the two domains. That is left to the reader to figure out. I suggest that the author provides some more guidance to the reader in this regard.
Please see the last paragraph in section 3 which addresses the comment from the reviewer
"The principles of the core ethical guidelines of the ACA, APA, AMA, and NASW namely autonomy and informed consent; beneficence and non-malfeasance; confidentiality, privacy and transparency; justice, fairness and inclusiveness; and fidelity, professional integrity and accountability and the ethical guidelines for AI by multinational technology corporations, and government and non-government stakeholders were examined for convergent principles with the purpose of informing recommendations for a comprehensive set of integrated ethical guidelines for mental health professionals in the face of the rapid expansion of generative AI in healthcare and specifically for mental health professionals" .
Reviewer Comment #6
Lines 267–270:
The author writes: “Secondly, while efforts in this paper were to consolidate human related ethical guidelines with machine related ethical guidelines, this amalgamation may be fraught with practical real-world implications and applications.”
Here one might wonder. What, more precisely, might this be? I think the paper would benefit from a bit more clarity here.
Clarity is offered through rewriting this section in the limitations with a specific example
Lines 274–275:
The author writes: “The proposed framework has not been empirically tested and validated in clinical settings”
This statement is interesting. What would such a validation look like? The framework consists (mostly) of a series of imperatives – how does the author conceive of them being validated? A few words on this might shed light on the matter.
Thank you. The limitation section was revised to reflect what validation will look like.
Reviewer 2 Report
Comments and Suggestions for Authors
The paper presents an important and timely discussion on ethical decision-making guidelines for mental health professionals in the era of artificial intelligence. It is commendable that the author attempts to synthesize existing ethical codes with emerging AI principles. However, there are areas where the paper can be strengthened both conceptually and structurally.
First, while the introduction provides a broad historical background, some parts read more like a general narrative of AI rather than a focused entry into the ethical challenges in mental health. I recommend streamlining this section to foreground the specific gap in ethical decision-making for clinicians earlier, rather than devoting lengthy space to AI history or user statistics. This would make the argument sharper and more relevant to the target readership.
Second, the framework is well-organized into five pillars, but it remains highly descriptive. The paper would benefit from deeper critical analysis that illustrates how these pillars can be applied in practice. For example, case-based examples (real or hypothetical) would help demonstrate the practical dilemmas clinicians might face and how the framework guides resolution. This would make the framework more usable and grounded.
Third, there is an over-reliance on American professional bodies and Western-centric AI ethics guidelines. While these are foundational, a more global perspective—including voices from low- and middle-income countries—would add value, given the international readership of the journal. The inclusion of culturally diverse ethical considerations would enrich the universality of the proposed framework.
Fourth, the methodology of how sources were reviewed is not clearly explained. At present, it appears as a selective synthesis rather than a systematic mapping of ethical codes and AI guidelines. A brief explanation of the process (e.g., inclusion criteria, scope of guidelines reviewed) would increase the rigor and transparency of the work.
Fifth, in terms of citations, the manuscript draws on organizational codes but misses engagement with more recent empirical or conceptual research in the education and healthcare AI literature. I recommend the author to consider to include the following relevant publications.
Sixth, the limitations section is honest but underdeveloped. The paper could expand on the implications of these limitations, particularly in relation to the urgent need for empirical validation of ethical frameworks. Suggesting possible methodologies for testing (such as surveys of clinicians or Delphi studies with experts) would make the contribution more forward-looking.
Finally, stylistically, the manuscript would benefit from clearer transitions between sections. At times, the flow feels like a list of codes and principles without adequate synthesis. Stronger linking sentences and critical comparisons across guidelines would help the reader follow the argument more coherently.
In summary, the manuscript offers a useful preliminary framework but needs more critical depth, global inclusivity, methodological transparency, and engagement with relevant scholarly works. With these improvements, the paper could make a stronger and more lasting contribution to the literature on ethics, AI, and mental health practice.
Author Response
Reviewer Comments With My Responses in Bold Italics
The paper presents an important and timely discussion on ethical decision-making guidelines for mental health professionals in the era of artificial intelligence. It is commendable that the author attempts to synthesize existing ethical codes with emerging AI principles. However, there are areas where the paper can be strengthened both conceptually and structurally.
First, while the introduction provides a broad historical background, some parts read more like a general narrative of AI rather than a focused entry into the ethical challenges in mental health. I recommend streamlining this section to foreground the specific gap in ethical decision-making for clinicians earlier, rather than devoting lengthy space to AI history or user statistics. This would make the argument sharper and more relevant to the target readership.
Thank you--A paragraph was added to the historical introduction that addresses the ethical decision making for clinicians earlier in the manuscript-- see highlighted section. This section was consolidated to make the argument stronger and relevant to mental health professionals.
Second, the framework is well-organized into five pillars, but it remains highly descriptive. The paper would benefit from deeper critical analysis that illustrates how these pillars can be applied in practice. For example, case-based examples (real or hypothetical) would help demonstrate the practical dilemmas clinicians might face and how the framework guides resolution. This would make the framework more usable and grounded.
Based on this recommendation, a hypothetical case study has been added (see section 5) that clearly addresses how each of the 5 pillars can be applied to the clinical setting and the identified client. Section 6 was added to provide a checklist that clinicians can consult prior to incorporating AI into their clinical practice repertoire to short-circuit practical dilemmas.
Third, there is an over-reliance on American professional bodies and Western-centric AI ethics guidelines. While these are foundational, a more global perspective—including voices from low- and middle-income countries—would add value, given the international readership of the journal. The inclusion of culturally diverse ethical considerations would enrich the universality of the proposed framework.
This is an excellent point-- it is added to the limitation of the scope of the literature search that was conducted. The author focused on professional codes of history that had a long track record as far back as the 1950s and have stood the test of time and the scrutiny of scholars- but the point is well taken -- to this end the case study that has been added includes a scenario that includes a diversity as it relates to applying the 5 pillar study structure to clinician and the international client with services provided by an international clinician. (see section 5).
Fourth, the methodology of how sources were reviewed is not clearly explained. At present, it appears as a selective synthesis rather than a systematic mapping of ethical codes and AI guidelines. A brief explanation of the process (e.g., inclusion criteria, scope of guidelines reviewed) would increase the rigor and transparency of the work.
The methodology section has been revised to provide more details about the process that was followed using the narrative review methodology. Citations from the literature that were used are also added
Fifth, in terms of citations, the manuscript draws on organizational codes but misses engagement with more recent empirical or conceptual research in the education and healthcare AI literature. I recommend the author to consider to include the following relevant publications.
Thank you -- 5 additional peer reviewed citations were added in text and in the references
Sixth, the limitations section is honest but underdeveloped. The paper could expand on the implications of these limitations, particularly in relation to the urgent need for empirical validation of ethical frameworks. Suggesting possible methodologies for testing (such as surveys of clinicians or Delphi studies with experts) would make the contribution more forward-looking.
Thank you for this feedback. The heading for section 6 is now Limitations and Future Directions and contains recommendations for how the limitations can be addressed by future practices, research and empirical validation.
Finally, stylistically, the manuscript would benefit from clearer transitions between sections. At times, the flow feels like a list of codes and principles without adequate synthesis. Stronger linking sentences and critical comparisons across guidelines would help the reader follow the argument more coherently.
Thank you -- transitional sentences were added between sections.
In summary, the manuscript offers a useful preliminary framework but needs more critical depth, global inclusivity, methodological transparency, and engagement with relevant scholarly works. With these improvements, the paper could make a stronger and more lasting contribution to the literature on ethics, AI, and mental health practice.
Thank you for the comprehensive review
Reviewer 3 Report
Comments and Suggestions for Authors
The paper addresses an interesting topic — the ethical decision-making challenges faced by mental health professionals in the age of AI. It summarizes well major ethical principles from mental health associations and AI governance frameworks, offering readers a well-organized synthesis around five clear pillars. But, the manuscript remains largely descriptive and lacks scientific depth or methodological transparency. It would benefit from clarifying how the integration between human ethics and AI ethics was operationalized (e.g., selection criteria, analytical framework). The proposed model could be strengthened through illustrative cases, empirical validation, or cross-disciplinary consultation. I'd rather say the text is well-written, but its contribution would be greater if it moved beyond summarizing existing codes toward generating new conceptual or practical insights for professional practice.
Author Response
Reviewer Comment:
The paper addresses an interesting topic — the ethical decision-making challenges faced by mental health professionals in the age of AI. It summarizes well major ethical principles from mental health associations and AI governance frameworks, offering readers a well-organized synthesis around five clear pillars. But, the manuscript remains largely descriptive and lacks scientific depth or methodological transparency. It would benefit from clarifying how the integration between human ethics and AI ethics was operationalized (e.g., selection criteria, analytical framework). The proposed model could be strengthened through illustrative cases, empirical validation, or cross-disciplinary consultation. I'd rather say the text is well-written, but its contribution would be greater if it moved beyond summarizing existing codes toward generating new conceptual or practical insights for professional practice.
Response:
Thank you for your feedback.
1.The methods section has been revised to address the reviewers concern about 'methodological' transparency.
2. A case study has been added to take the synthesis and descriptive focus of the 5 pillars to demonstrate how the ethical framework could be applicable to diverse mental health clinicians serving diverse mental health clients.
3. In addition, a section has been added to the manuscript " Checklist for Clinicians Prior to AI Use" that provides practical guidelines that a mental health clinician can use.
4. Suggestions for empirical validation is offered in the limitations section
Round 2
Reviewer 3 Report
Comments and Suggestions for Authors
The manuscript addresses an important and timely topic at the intersection of mental health practice and AI ethics. It offers a clear and well-organized synthesis of ethical principles drawn from both traditional mental health codes and emerging AI guidelines. The writing is clear, and the proposed framework is easy to follow. At the same time, the methodological description of the narrative review could be strengthened.The methodology section would benefit from additional detail on the search strategy, inclusion criteria, and synthesis process to improve transparency and strengthen the scientific rigor of the narrative review. Clarifying how the final ethical framework was derived from the reviewed sources would further enhance the credibility of the proposed guidelines.
Author Response
Reviewer Comment:
"At the same time, the methodological description of the narrative review could be strengthened. The methodology section would benefit from additional detail on the search strategy, inclusion criteria, and synthesis process to improve transparency and strengthen the scientific rigor of the narrative review. Clarifying how the final ethical framework was derived from the reviewed sources would further enhance the credibility of the proposed guidelines."
Author Response: Thank you for the additional feedback.
I have added additional information that I previously deleted ( because of the brief report category) based on the reviewers most recent recommendation. I have highlighted the edits in the methodology section and throughout the manuscript. I have also revised the order of the references in the reference list and made the necessary edits in the body of the manuscript.